# Two-Stage Multi-Scale Fault Diagnosis Method for Rolling Bearings with Imbalanced Data

**Minglei Zheng** [1], **Qi Chang** [1], **Junfeng Man** [1,2,*], **Yi Liu** [3] **and Yiping Shen** [4]

[1] School of Computers, Hunan University of Technology, Zhuzhou 412007, China; m20085400011@stu.hut.edu.cn (M.Z.); m19085208007@stu.hut.edu.cn (Q.C.)
[2] School of Computers, Hunan First Normal University, Changsha 410205, China
[3] National Innovation Center of Advanced Rail Transit Equipment, Zhuzhou 412007, China; liy@chinazrcc.com
[4] Hunan Provincial Key Laboratory of Health Maintenance for Mechanical Equipment, Hunan University of Science and Technology, Xiangtan 411201, China; ypsh@hnust.edu.cn
* Correspondence: manjunfeng@hut.edu.cn

**Abstract:** Intelligent bearing fault diagnosis is a necessary approach to ensure the stable operation of rotating machinery. However, it is usually difficult to collect fault data under actual working conditions, leading to a serious imbalance in training datasets, thus reducing the effectiveness of data-driven diagnostic methods. During the stage of data augmentation, a multi-scale progressive generative adversarial network (MS-PGAN) is used to learn the distribution mapping relationship from normal samples to fault samples with transfer learning, which stably generates fault samples at different scales for dataset augmentation through progressive adversarial training. During the stage of fault diagnosis, the MACNN-BiLSTM method is proposed, based on a multi-scale attention fusion mechanism that can adaptively fuse the local frequency features and global timing features extracted from the input signals of multiple scales to achieve fault diagnosis. Using the UConn and CWRU datasets, the proposed method achieves higher fault diagnosis accuracy than is achieved by several comparative methods on data augmentation and fault diagnosis. Experimental results demonstrate that the proposed method can stably generate high-quality spectrum signals and extract multi-scale features, with better classification accuracy, robustness, and generalization.

**Keywords:** imbalanced data; bearing fault diagnosis; multi-scale; generative adversarial networks





## 1. Introduction

Rolling bearings are some of the most easily damaged components in rotating machinery, especially when under long-term high-speed and heavy load operation; inner ring, outer ring, and ball faults occur frequently [1]. Therefore, research into fault diagnosis methods of rolling bearings has practical engineering significance and economic value and is a hot spot in the field of mechanical fault diagnosis. According to relevant statistics, about 40% of rotating machinery faults are caused by problems with the bearings [2,3]. Therefore, research on bearing fault diagnosis methods is of great significance to the safe operation of rotating machinery, which can avoid huge economic losses and casualties as a result of accidents [4]. In recent years, data-driven fault diagnosis methods have gradually become a research hotspot in the field of intelligent machine fault diagnosis [5–7]. Tang et al. [8] proposed a method that combined variational mode decomposition (VMD) and support vector machine (SVM) to decompose the original signal into several intrinsic mode functions (IMFs) using the time-frequency analysis method of VMD, which can filter the noise component with the kurtosis index, and input the filtered IMFs as fault features to the SVM for fault diagnosis. Lei [9] proposed an unsupervised pre-training of deep neural network (DNN) methods by using a stack auto-encoder with tie-weights, followed by the fine-tuning of a pre-training DNN using a back propagation (BP) algorithm under

supervised conditions. The pre-training process can help to mine fault features and the fine-tuning process helps to obtain discriminant features for fault classification. Levent et al. [10] proposed a deep learning algorithm that used a one-dimensional convolutional neural network (CNN) to extract the local features of time-series signals for fault diagnosis. Duan et al. [11] proposed a method, based on ResNet as the backbone network, to improve the classification accuracy of fault diagnosis, which can be used to layer feature maps, compare potential candidate model blocks, and screen out the best feature combinations. Rhanoui et al. [12] developed a bidirectional long short-term memory network (BiLSTM)-based method for bidirectional feature extraction using forward and reverse position sequences of frequency-domain signal data, which can extract better sequence feature information than unidirectional LSTM [13]. Huang et al. [14] proposed a method combining CNN and LSTM, which used a convolutional layer to extract feature information and LSTM to extract timing information. Jiao et al. [15] designed a CNN-LSTM network based on an attention mechanism, which used a hierarchical attention mechanism to establish the importance of the extracted features, thus enhancing the performance and interpretability of the neural network learning process. It is worth noting that the attention model (AM) is widely used in feature extraction, which can calculate the importance of different features. Therefore, we decided to introduce a multi-scale mechanism to establish the features of vibration signals at different scales, so that we could significantly improve the robustness of the model under different working conditions, which is conducive to the application of the model [16].

Data-driven intelligent diagnostic models always have a huge demand for sufficiently labeled training data under all health conditions. However, in actual working conditions, the fault samples of mechanical equipment collected by sensors are generally far less accessible than normal samples, which leads to a serious imbalance problem with the dataset for fault diagnosis [17]. In the process of continuous training, data-driven models always tend to improve the classification accuracy of more classes of samples and drop the classification performance level on a few classes of samples, which causes severe challenges for fault diagnosis. At present, the oversampling method is mostly used to deal with the problem of imbalanced data, a method that can artificially generate new data to increase the amount of data available. One of the most widely used methods for data augmentation is the synthetic minority oversampling technique (SMOTE) [18] and its improved method [19], which generates new samples through the interpolation of real data. However, the data generated by SMOTE based on the k-nearest neighbor principle still obey an uneven minority sample distribution, which means that SMOTE cannot really change the distribution of the dataset. It is easy to produce a boundary effect, resulting in the fuzziness of the classification boundary of adjacent categories. The other data augmentation method is with a generative adversarial network (GAN) [20].

After the GAN model was proposed, it was at the forefront of the trend in data augmentation. Many scholars have proposed methods to solve the imbalanced data problem by using a GAN model [21,22] in different fields [23–25], such as images, audio, text, fault diagnosis, etc. Lee et al. [26] studied and compared the GAN-based oversampling method and standard oversampling method in terms of the imbalanced data of the fault diagnosis of electric machines, then combined it with the deep neural network model; they proved that the GAN model generated data with higher classification accuracy. However, GAN still has many drawbacks, such as training instability, training failure, vanishing gradients, and mode collapse. To solve these problems, researchers have proposed a deep convolution network based on GAN (DCGAN) [27], a WGAN [28] using Wasserstein distance instead of Jensen–Shannon dispersion, and WGAN-GP [29] using the gradient penalty. These GAN methods improved the model, in terms of the network structure and loss function, to make the quality of generated data better and the training process more stable. Shao et al. [30] used this improved framework of DCGAN to learn the original vibration data collected by sensors on machines, which generated one-dimensional signal data to alleviate the data imbalance. Zhang et al. [31] proposed a method to generate a few categories of EEG samples based on a conditional Wasserstein GAN, which enhanced

the diversity of generated time-series samples. Li et al. [32] proposed a fault diagnosis method for rotating machinery based on an auxiliary classifier and Wasserstein GAN with a gradient penalty, which improved the validity of the generated samples and the accuracy of fault diagnosis. Zhou et al. [33] proposed an improved GAN method that uses an autoencoder to extract fault features, which is used to generate fault features instead of using real fault data samples, but this method will inevitably mean losing the physical information of signals. Zhang et al. [34] designed a DCGAN structure to learn the mapping relationship between noise and mechanical vibration data; however, this complex structure makes the model unstable and easy to lead to training failure, which needs to be improved.

In summary, the existing improved GAN generation methods have made breakthroughs in the design of network structure and loss function. However, in the experiment based on a one-dimensional vibration signal, there are still some problems, such as the instability of model training and the damage to signal mechanism information. Therefore, based on the idea of GAN variants such as DCGAN [27], WGAN-GP [29], StackGAN [35], and ProGAN [36], and the multi-scale mechanism, a two-stage bearing fault diagnosis method for imbalanced data is proposed. This includes a multi-scale progressive generative adversarial network, named MS-PGAN, for data augmentation and a MACNN-BiLSTM, based on the multi-scale attention fusion mechanism, for bearing fault diagnosis. This framework has the advantages of high stability, fast convergence, and strong robustness. The contributions of this paper are summarized as follows:

- Stage 1: A multiscale progressive generative adversarial network is proposed, to generate high-quality multi-scale data to rebalance the imbalanced datasets.
  - a. A multi-scale GAN network structure with progressive growth has strong stability, which avoids the common problem of training failure in the GAN.
  - b. The improved loss function MMD-WGP makes the generator model learn the distribution of fault samples from normal samples by introducing the transfer learning mechanism [37], which effectively improves the problem of random spectral noise and mode collapse.
  - c. The local noise interpolation upsampling uses adaptive noise interpolation in the process of dimension promotion to protect the frequency information of the fault feature.

- Stage 2: Combined with multi-scale MS-PGAN, a diagnostic method based on a multi-scale attention fusion mechanism, named MACNN-BiLSTM, is proposed.
  - a. The feature extraction structure of the proposed diagnosis method can combine the local feature extraction capability of the CNN and the global timing feature extraction capability of BiLSTM.
  - b. The multi-scale attention fusion mechanism enables the model to fuse feature information extracted from different scales, which significantly improves the diagnostic capability of the model.

The remainder of the paper is organized as follows. The theoretical background is presented in Section 2. The proposed methodology and detailed framework are described in Section 3. Experimental details, results, and our analysis are presented in Section 4. Finally, our conclusions are drawn in Section 5.

## 2. Theoretical Background

### 2.1. Convolutional Neural Network (CNN)

Deep learning has a deeper network layer and a larger hierarchical structure than shallow neural networks. It mainly obtains a deep abstract expression by extracting and combining the underlying features. Among them, the recurrent neural network (RNN), convolutional neural networks (CNN), and various variant models are the most widely used, and the actual effect is the most ideal [38]. CNN is a feed-forward neural network with convolution computation and a deep structure, which is composed of a convolution layer, activation layer, pooling layer, full connection layer, etc. The CNN uses a convolution

kernel to simulate the function of the human visual cortex receptive field to extract a feature map. The calculation formula of convolution is expressed as:

$$y_i^{l+1} = f\left(W_i^{l+1} * X^l + b_i^{l+1}\right) \tag{1}$$

where $y_i^{l+1}$ is the *i*th value of the output of the $l+1$th layer, $f(\cdot)$ is the activation function, $W_i^{l+1}$ is the shared weight of the *i*th convolution kernel in the $l+1$th convolution layer, $X^l$ is the output of the *l*th layer and the input of the $l+1$th layer, and $b_i^{l+1}$ is the *i*th bias of the $l+1$th layer.

The activation function provides the CNN with the ability to solve nonlinear problems. Common activation functions include Sigmoid, Tanh, ReLU, LeakyReLU, etc. These activation functions have their own advantages and disadvantages, so we can choose among them flexibly according to the actual network situation. The pooling layer is used to reduce network parameters and avoid overfitting. There are two main types of pooling: maximum pooling and average pooling. Currently, maximum pooling is widely used because it can save the most important information in the pooling window and avoid feature blurring. The function of the full connection layer is to combine the extracted features in a nonlinear way to obtain the output. Some improved networks use global average pooling instead of using the full connection layer.

### 2.2. Generative Adversarial Network (GAN)

The GAN [20] is a generative model of adversarial deep learning. It generates data through the interplay of the generator and discriminator. Its core idea comes from the concept of "Nash equilibrium" in game theory [39]. The GAN learns the data distribution of training samples in the game process of generating a network and identifying a network, then generates new data similar to the original data distribution to achieve the effect of data enhancement. Therefore, the generated network output can be used to confuse the training samples with the real samples, so as to solve the problem of imbalanced data, that the fault samples are less than the normal samples in the actual fault diagnosis. As shown in Figure 1, GAN consists of two different sub-networks, the generator and the discriminator, which are trained at the same time.

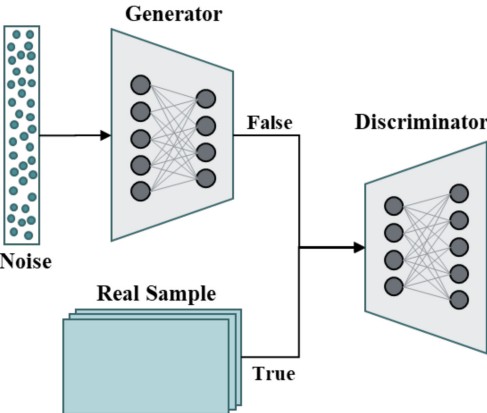

**Figure 1.** The structure of the GAN model.

The generator inputs a set of random noise $Z = \{z_1, z_2, \cdots, z_m\}$, the generated false sample $G(z) = \{G(z_1), G(z_2), \cdots, G(z_m)\}$, which is the same as the real data dimension; the discriminator is responsible for distinguishing the real sample $X = \{x_1, x_2, \cdots, x_n\}$, and the generated false sample $G(z) = \{G(z_1), G(z_2), \cdots, G(z_m)\}$. The loss function of GAN is expressed as:

$$\min_G \max_D V(D, G) = E_{x \sim P_{data}(x)}[\log D(x)] + E_{z \sim P_z(z)}[\log(1 - D(G(z)))] \tag{2}$$

where $P_{data}(x)$ is the data distribution of real data, $P_z(z)$ is the a priori noise distribution. $D(x)$ represents the probability that x comes from real data. $D(G(z))$ represents the probability that $G(z)$ comes from the generated data, where $G(z)$ is the data sample generated by the generator from the noise data $z$, subject to the a priori distribution. $E_{x \sim P_{data}(x)}$ represents the data distribution expectation of $x$ from real data, while $E_{z \sim P_z(z)}$ represents the expectation that $z$ comes from the noise distribution.

## 3. Proposed Methodology

In this section, the proposed two-stage method for bearing fault diagnosis is described. Figure 2 is a flowchart of the two-stage approach presented in this paper. The two-stage method consists of data augmentation and fault diagnosis. In stage 1, to solve the problem of imbalanced data, MS-PGAN was used to generate data and expand the dataset. In stage 2, the multi-scale data generated by MS-PGAN are input into the MACNN-BiLSTM model, then the multi-scale feature information is extracted and fused for fault classification.

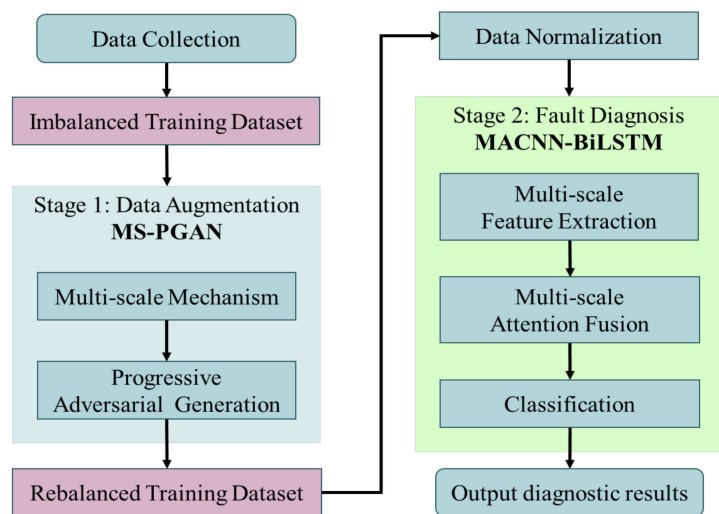

**Figure 2.** The flowchart of the proposed two-stage approach.

### 3.1. MS-PGAN

3.1.1. The Structure of MS-PGAN

The MS-PGAN presented in this paper is a stable and fast-convergent multi-scale progressive GAN framework. Figure 3 clarifies the network structure of MS-PGAN, which includes an input unit, multi-layer GAN sub-structure, and upsampling unit. Each GAN substructure is based on a convolutional GAN, including a generator and discriminator. The parameters of the structure are shown in Tables 1 and 2. The generator has a 5-layer network; the activation function of the middle layer uses ReLU, and the activation function of the output layer uses Tanh. The discriminator has a 5-layer network, with LeakyReLU as the activation function for the middle layer and softmax as the activation function for the output layer. B is the batch size, and N is the input scale size. Convolution in deep learning is generally used to process two-dimensional images. To process one-dimensional signal data, we use 1D-Conv and 1D-ConvT. The name "1D-Conv" refers to a one-dimensional convolution layer, which is used to extract and compress the one-dimensional input features. Conversely, 1D-ConvT refers to a one-dimensional transposed convolution layer, which is used to amplify the length of the one-dimensional input data. It works almost exactly the same as the one-dimensional convolutional layer but in reverse. The input size parameter N of the GAN sub-networks with different dimensions is determined by the dimension of the input data. The distribution of the input data of each layer is transformed into normal distribution by using batch normalization, which causes the mean and variance of input data to remain standardized to reduce the shift in internal covariance, so as to alleviate the

problem of gradient disappearance and accelerate the convergence of the model. The input unit and upsampling unit are described in detail in a later section.

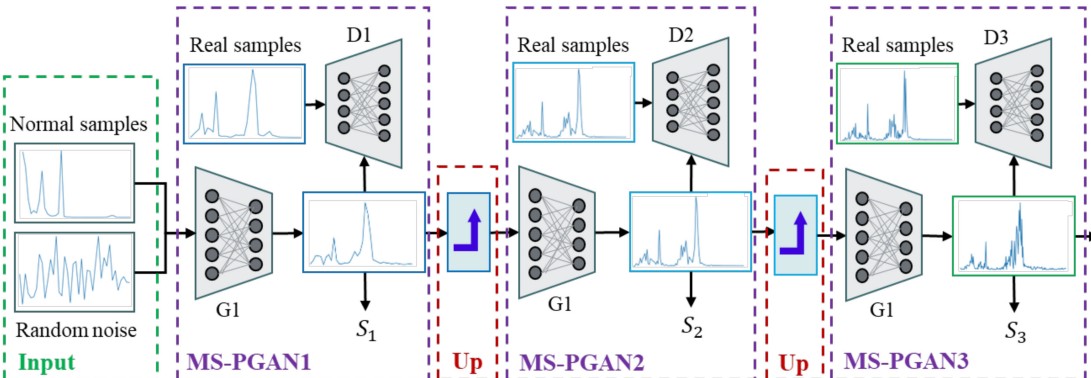

**Figure 3.** The structure of the MS-PGAN model.

**Table 1. The** MS-PGAN's generator network structure parameters.

| Layer | Input Size | Output Size | BN | Activation Function | Layer |
|---|---|---|---|---|---|
| 1D-ConvT | B,N,1,1 | B,64 * 2,1,5 | yes | ReLU | 1D-ConvT |
| 1D-ConvT | B,64 * 2,1,5 | B,64 * 4,1,12 | yes | ReLU | 1D-ConvT |
| 1D-ConvT | B,64 * 4,1,12 | B,64 * 2,1,24 | yes | ReLU | 1D-ConvT |
| 1D-ConvT | B,64 * 2,1,24 | B,64,1,50 | yes | ReLU | 1D-ConvT |
| 1D-ConvT | B,64,1,50 | B,1,1,N | yes | Tanh | 1D-ConvT |

**Table 2.** MS-PGAN's discriminator network structure parameters.

| Layer | Input Size | Output Size | BN | Activation Function | Layer |
|---|---|---|---|---|---|
| 1D-Conv | B,1,1,N | B,64,1,50 | yes | LeakyReLU | 1D-Conv |
| 1D-Conv | B,64,1,50 | B,64 * 2,1,24 | yes | LeakyReLU | 1D-Conv |
| 1D-Conv | B,64 * 2,1,24 | B,64 * 4,1,12 | yes | LeakyReLU | 1D-Conv |
| 1D-Conv | B,64 * 4,1,12 | B,64 * 2,1,5 | yes | LeakyReLU | 1D-Conv |
| 1D-Conv | B,64 * 2,1,5 | B,1,1,1 | yes | Softmax | 1D-Conv |

The training process of MS-PGAN is an adversarial process of progressive growth. The algorithm procedure of MS-PGAN is shown in Algorithm 1. Firstly, the training generator $G_1$ achieves Nash equilibrium with the discriminator $D_1$ by inputting a low-scale normal frequency spectrum signal $X_{Normal}$ and a low-scale real sample $S_1^{real}$ with Gauss noise $Z$ into the generator $G_1$ and generates the same-scale fault spectrum signal sample, $S_1$, for the fault class. The generated low-scale fault samples are transformed into the mesoscale fault sample $S_1'$ by local noise interpolation upsampling, then the $S_1'$ and the real sample $S_2^{real}$ are input into the generator. After the Nash balance between the training generator $G_2$ and the discriminator $D_2$ is reached, a specified number of mesoscale fault frequency spectrum samples $S_2$ are generated. The same method is used to generate a high-scale spectrum signal $S_3$ or even higher-scale samples. Finally, the generated samples of the spectrum signal with different scales are obtained from each GAN subnetwork.

---

**Algorithm 1.** The procedure of MS-PGAN

---

**Input:** $X_{Normal}, Z, S_1^{real}, S_2^{real}, S_3^{real}$
**Output:** $S_1, S_2, S_3$
1:   **for** $i = 1$ to 3 **do**
2:       **if** $k_i$ is 40:
3:           $G_1, D_1 \ X_{Normal}, Z, S_1^{real}$
4:           **while** $G_1, D_1$ Satisfy Nash equilibrium **do**
5:               $G_1, D_1 \leftarrow \min_D\max_G x L_{MS-PGAN}(D_1, G_1)$
6:               $S_1 \leftarrow G_1(X_{Normal} + Z)$
7:           **end while**
8:       **end if**
9:   **if** $k_i$ is 100 or 200:
10:           $S'_{i-1} \leftarrow \text{upsampling}(S_{i-1})$
11:           $G_i, D_i \leftarrow S'_{i-1}, S_i^{real},$
12:           **while** $G_i, D_i$ Satisfy Nash equilibrium **do**
13:               $G_i, D_i \leftarrow \min_D\max_G x L_{MS-PGAN}(D_i, G_i)$
14:               $S_i \leftarrow G_1\left(S'_{i-1}\right)$
15:           **end while**
16:       **end if**
17: **end for**

---

### 3.1.2. Multi-Scale Mechanism

The multi-scale mechanism can transform time-series signals into multi-scale one-dimensional signals. Due to the different levels of features of signals that can be observed at different scales, a multi-scale mechanism has the ability to better represent the feature information and improve the performance of the network. In this paper, a multi-scale processing mechanism has been designed to process the data in the frequency domain of the vibration signals. The 1D maximum pool method was selected, and its step size is consistent with the kernel size. The maximum value in the sliding window can better preserve the instantaneous energy features of the signal and filter out random noise and high-frequency disturbance to a certain extent. In the training process, a set of vibration signals $\{x_1, \cdots, x_n, \cdots, x_N\}$ is used, where $N$ is the length of the original input data and $x_n$ is the $n$th vibration value of the original input signal. The corresponding scale time series are obtained by one-dimensional maximum pooling with different steps, and the final data length is $\frac{N}{s}$. The multi-scale calculation process is expressed as:

$$M_{s,j} = \max\left\{X'_j\right\}, 1 \leq j \leq \frac{N}{S} \tag{3}$$

where $M_{s,j}$ is an output signal obtained for multi-scale processing. $M$ is the step size of one-dimensional max pooling, while $X'_j$ is the $j$th frequency spectrum signal sequence with scale $S$. Multi-scale fault category frequency spectrum signals are divided into three scale types:

- Low-dimensional rough scale: if $N$ is 200 and $S$ is 5 and the resolution is 40-length, which mainly includes the features of spectral peak.
- Middle-dimensional scale: if $N$ is 200, $S$ is 2 and the resolution is 100-length, so more harmonic features are added.
- High dimensional scale: if $N$ is 200, $S$ is 1 and the resolution is 200-length; this enriches the detailed features of the signal, including the complete frequency domain information.

### 3.1.3. Improved GAN Loss Function with Transfer Learning

WGAN proposes the measurement of Wasserstein distance to replace the JS divergence of the basic Gan model, keeps the loss function of the traditional Gan unchanged, removes the sigmoid layer of the discriminator (D) loss, and cancels the logarithmic process of the

generator (*G*) and discriminator (*D*) loss. In addition, because the Wasserstein distance has no upper and lower bounds, it may cause *D* to become larger and larger after multiple iterations and updates. Therefore, WGAN limits the absolute value of discriminator *D* to a constant "C" by weight-clipping every time it updates the discriminator *D*, which makes the value of *D* smoother, and effectively alleviates the difficulties and instability of GAN training due to loss of function, mode collapse, and other problems. However, the trick used in WGAN does not really make *D* satisfy the requirement that for any x, the magnitude of the gradient is less than or equal to 1, to meet the "1-Lipschitz" condition. Therefore, problems such as training difficulties and slow convergence will still be encountered in practical applications. Therefore, an improved method, WGAN-GP, was proposed. WGAN-GP realizes the approximate "1-Lipschitz" condition restriction on discriminator *D* by using a gradient penalty instead of the weight-clipping method of directly clipping the gradient value. This method has achieved good performance in practice. The actual gradient penalty term of WGAN-GP is shown in Formula (4). The original loss function of the discriminator is shown in Formula (5), and the loss function of the discriminator of WGAN-GP is shown in Formula (6).

$$GP = E_{x \sim P_{Penalty}}\left[(||\nabla_x D(x)||_2 - 1)^2\right] \tag{4}$$

$$L_{WGAN}(G, D) = E_{x \sim P_G}[D(x)] - E_{x \sim P_{data}}[D(x)] \tag{5}$$

$$L(D) = L_{WGAN} + GP \tag{6}$$

Although the loss function proposed by WGAN-GP has been successful in the field of GAN image generation, it has been found in experiments that there are still problems regarding the generation of a one-dimensional vibration signal, such as random spectral noise and mode collapse.

In the case of sparse and limited training data, transfer learning [37] can build a powerful generalization model from related but different application scenarios to create new application scenarios by making the source domain instance distribution close to the target domain instance distribution. From the perspective of transfer learning, the essence of a fault signal is a normal signal with added fault features. This means that the domain distribution of normal signals from rotating machinery equipment is related to but different from that of fault signals, which meets the application conditions of transfer learning. Therefore, we introduced a transfer learning mechanism to improve the model. Taking normal samples as the source domain $D_X$ and fault samples as the target domain $D_Y$, the MS-PGAN model is trained to learn the distribution of fault samples from a sufficient number of normal samples. The process of transfer learning is shown in Figure 4. Its advantages are as follows: (1) learning the frequency features of fault samples from the frequency features of normal samples can significantly reduce the random spectral noise in the generated samples, which can retain the original physical information to the maximum extent; (2) it avoids the generator directly learning features from a small amount of fault data, causing the problem of mode collapse.

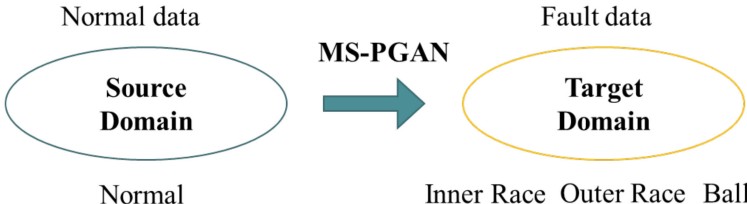

**Figure 4.** The process of transfer learning.

The proposed method presents MMD-WGP as a loss function of the MS-PGAN model. On the basis of WGAN-GP, the maximum mean discrepancy (MMD) [40,41], which measures the similarity between source domain and target domain in the transfer learning domain, is introduced to measure the similarity between generated samples and real sam-

ples. In the experiment, using the WGAN-GP loss function in a one-dimensional spectrum signal generation task, the model converges too fast in a vanishing gradient, which leads to inadequate training and so the model is difficult to optimize. The suggested method solves this problem by introducing the MMD penalty of maximum mean difference, which makes the model training more stable and generates samples closer to the true fault signal. MMD is expressed as:

$$MMD[F, p, q] = \sup_{f \in F} \left( E_{x \sim p}[f(x)] - E_{x \sim q}[f(x)] \right) \tag{7}$$

where $F$ denotes a given set of functions, $p$ and $q$ are two independent distributions, $x$ and $y$ obey $p$ and $q$, respectively, sup denotes an upper bound, and $f(\cdot)$ denotes a function mapping. We calculate the value of $MMD^2$ as the MMD penalty between source domain $D_X$ and the target domain $D_Y$, which is shown in Formula (8). The improved loss function MMD-WGP of the MS-PGAN model is shown in Formula (9).

$$MMD^2[D_X, D_Y] = \| \frac{1}{x} \sum_{i=1}^{x} f(x_i) - \frac{1}{y} \sum_{j=1}^{y} f(y_i) \|^2 \tag{8}$$

$$L_{MSGAN}(G, D) = E_{x \sim P_G}[D(x)] \quad -E_{x \sim P_{data}}[D(x)] + \lambda E_{x \sim P_{Penalty}} \left[ (\| \nabla_x D(x) \|_2 - 1)^2 \right]$$
$$+\mu \| \frac{1}{x} \sum_{i=1}^{x} f(x_i) - \frac{1}{y} \sum_{j=1}^{y} f(y_i) \|^2 \tag{9}$$

In Equation (9), the first two are the Wasserstein distance of WGAN-GP and the gradient penalty. The last one is the MMD penalty, which measures the distribution of generated fault samples and the distribution of the real fault samples.

### 3.1.4. Local Noise Interpolation Upsampling

In the training process of MS-PGAN, the progressive growth of generated samples requires the use of the upsampling method; that is, the input of low-scale generated fault signal samples to a higher-scale signal needs to be processed by upsampling. The main methods are nearest-neighbor interpolation, bilinear interpolation, deconvolution, etc. These classical upsampling methods are well applied in the GANs. In ProGAN [36], a progressively growing generation model, the conversion of pictures from low-dimensional pixels, $4 * 4$, to higher-dimensional pixels, $8 * 8$, is achieved through nearest-neighbor interpolation. In its improved model, StyleGAN [42], the core structure synthesis network uses deconvolution to convert the generated low-resolution pictures into higher-resolution pictures to double the resolution.

In this paper, based on the features of one-dimensional spectral time-series signals and MS-PGAN networks using input noise control to generate sample diversity, a local noise interpolation sampling method is proposed: that is, inserting adaptive Gaussian noise between two points of a low-dimensional signal. The mean $\sigma_i$ and variance $\mu_i$ of noise are determined by the local values of the local window $i \in \{1, 2 \ldots, N/k\}$, where $N$ is the length of the input and $k$ is the size of the local window. The distribution of adaptive Gaussian noise is expressed as:

$$p_G(x, a, b, i) = \frac{1}{b\sigma_i \sqrt{2\pi}} e^{-\frac{(x - a\mu_i)^2}{2(b\sigma_i)^2}} \tag{10}$$

where parameters $a$ and $b$ are coefficients of the mean and variance of the signals in the local window.

### 3.2. MS-PGAN Combining MACNN-BiLSTM

In recent years, many scholars have combined GAN or its variant with the CNN in the field of data imbalance, to form a fault diagnosis model of a deep convolutional generative adversarial network. On the other hand, Hochreite et al. [13] proposed a long

short-term memory network (LSTM), which has certain information-mining capabilities in terms of long sequential data and is widely applied in various fields related to time series. A bidirectional long short-term memory network (BiLSTM), a variant of LSTM, can learn bidirectional temporal features from the time series to capture more sequence dependencies. Therefore, we designed the model to fuse the ideas of the CNN and BiLSTM: the CNN's short-sequence feature abstraction ability is used to extract local features, while the BiLSTM integrates the short-sequence local features to extract bidirectional global temporal features, effectively improving the feature extraction ability of the model. Furthermore, by introducing a multi-scale attention fusion mechanism (MSAFM), the diagnostic model can establish the importance of feature information at different scales to adaptively select the best feature combinations at different scales for weighted fusion.

As shown in Figure 5, the model is composed of a data augmentation module, a multi-scale feature extraction module, a multi-scale attention fusion module, and a classification module. The MS-PGAN data augmentation module is composed of a sub-network for progressive adversarial generation at multiple levels, which can be used to output multi-scale rebalance samples. The multi-scale feature extraction module is composed of three CNN-BiLSTM sub-networks for different scales, wherein convolution blocks extract local features and the BiLSTM extracts long-term temporal-dependent information. Each one-dimensional convolution block (1D-Conv Block) contains a one-dimensional convolution layer, a BN layer, and a LeakyReLU activation layer. The end of each sub-network uses global average pooling (GAP) to reduce the model parameters, which can improve training speed and reduce overfitting. The multi-scale attention fusion mechanism fuses different scale features using an adaptive attention weight. The classification module consists of the full connection layer and the softmax layer, which can output the probability of label classification. The parameters of the MACNN-BiLSTM are shown in Table 3, where B is the batch size, N is the input scale size, and Num is the number of label categories.

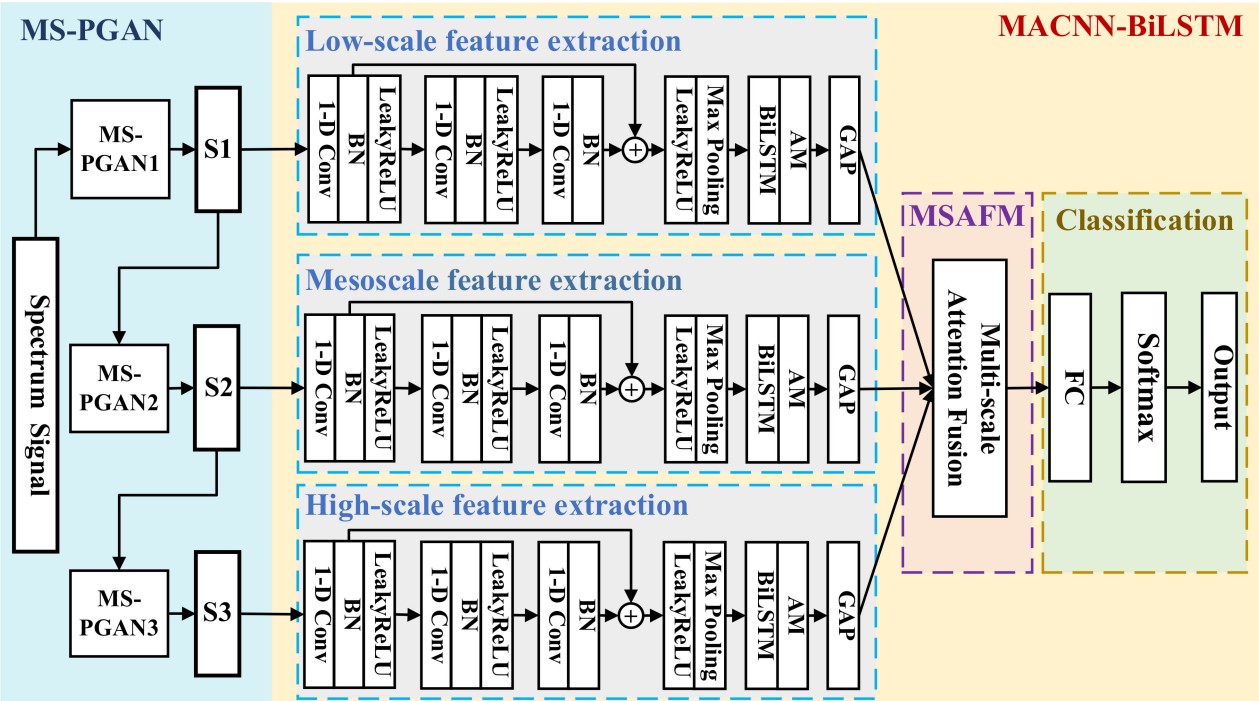

**Figure 5.** The network structure of MS-PGAN, combining the ACNN-BiLSTM.



**Table 3.** The parameters of the MACNN-BiLSTM.

| Layer | Input Size | Kernel Size | Stride | Padding |
|---|---|---|---|---|
| input | B,1, N | | | |
| 1D-Conv Block1 | B,128,N | 5,1 | 1,1 | yes |
| 1D-Conv Block2 | B,128,N | 5,1 | 1,1 | yes |
| 1D-Conv Block3 | B,128,N | 5,1 | 1,1 | yes |
| ADD | B,128,N | | | |
| LeakyReLU | B,128,N | | | |
| Max Pool | B,128,N | 2,1 | 1,1 | yes |
| BiLSTM | B,128,N/2 | | | |
| Attention | B,128,256 | | | |
| GAP | B,4,256 | 4,1 | 1,1 | yes |
| MS-Attention | B,256,1 | | | |
| FC | 96,Num | | | |
| softmax | Num,Num | | | |

Algorithm 2 is the procedure of MACNN-BiLSTM, the input of which is the multi-scale rebalancing training dataset $\{S_1, S_2, S_3\}$, which is the output of stage 1, and the output of that is the probability $O$ of fault classification. Firstly, $S_k$ at the scale of $k$ is input into the feature extraction module of the scale, then three continuous convolution blocks are used to extract the local features of $S_k$ to calculate $\{C_1^k, C_2^k, C_3^k\}$. Each convolution block contains a one-dimensional convolution layer, a BN layer, and a LeakyReLU activation layer. Then, a residual structure is used to input the sum of $C_3^k$ and $C_1^k$ into the LeakyReLU activation layer to prevent the vanishing gradient. The convolution layer output $C_o^k$ is obtained via the max-pooling layer, which is used to reduce the complexity of the feature maps and prevent the model overfitting. In addition, we input $C_o^k$ into the bidirectional LSTM network with care to establish the memory unit output, $M_o^k$. It can extract global temporal features, according to the weight of attention. Via the global average pooling layer at the end of each feature extraction subnetwork, the variable $H^k$ is output, reducing the number of model parameters to alleviate the overfitting problem. Furthermore, the multi-scale attention fusion mechanism is used to calculate the adaptive attention weight $a_{AM}$ of the outputs $\{H_1^k, H_2^k, H_3^k\}$ from different scales. Finally, the weighted fusion output $Y_{AM}$ is input into the full connection layer, and the classification probability $O$ of fault diagnosis is output via the softmax layer.

---

**Algorithm 2.** The procedure of MACNN-BiLSTM

---

**Input:** $S_1, S_2, S_3$
**Output:** $O$
1:   **while** not converge **do**
2:       **for all** $S_k$ **do** $S_k$
3:           $C_0^k = S_k$
4:           **for** $i$ = 1 to 3 **do**
5:               $C_i^k \leftarrow$ Conv Block$_i \left( C_{i-1}^k \right)$
6:           **end for**
7:           $C_o^k \leftarrow$ MP(LeakyReLU($C_3^k + C_1^k$))
8:           $M_o^k \leftarrow$ Attention(BiLSTM($C_o^k$))
9:           $H^k \leftarrow$ GMP($M_o^k$)
10:       **end for**
11:       $\alpha_{AM} \leftarrow$ Softmax(FC(Concat($H^1, H^2, H^3$)))
12:       $Y_{AM} \leftarrow \alpha_{AM} *$ Concat $(H^1, H^2, H^3)$
13:       $O \leftarrow$ Softmax(FC($Y_{AM}$))
14: **end while**

---

## 4. Experimental Study

In order to verify the validity, robustness, and generalization of the proposed model in solving the problem of an imbalanced sample of fault signals in actual conditions, two different datasets of rotating machinery are selected for the experimental study.

### 4.1. Dataset Descriptions and Preprocessing

#### 4.1.1. Case 1: UConn Dataset

The University of Connecticut (UConn) gearbox dataset is a gearbox vibration dataset shared by Professor Tang Liang's team [43]. As shown in Figure 6, the experimental equipment is a benchmark two-stage gearbox with replaceable gears. The speed of the gear is controlled by a motor and the torque is provided by a magnetic brake, which can be adjusted by changing its input voltage. The first input shaft has 32 pinions and 80 pinions, and the second stage consists of 48 pinions and 64 pinions. The input shaft speed is measured by a tachometer with teeth. The vibration signal is recorded by the dSPACE system with a sampling frequency of 20KHz. Nine different health states are introduced to the pinion on the input shaft, including healthy conditions, a missing tooth, a root crack, spalling, and a chipping tip, with five different levels of severity.

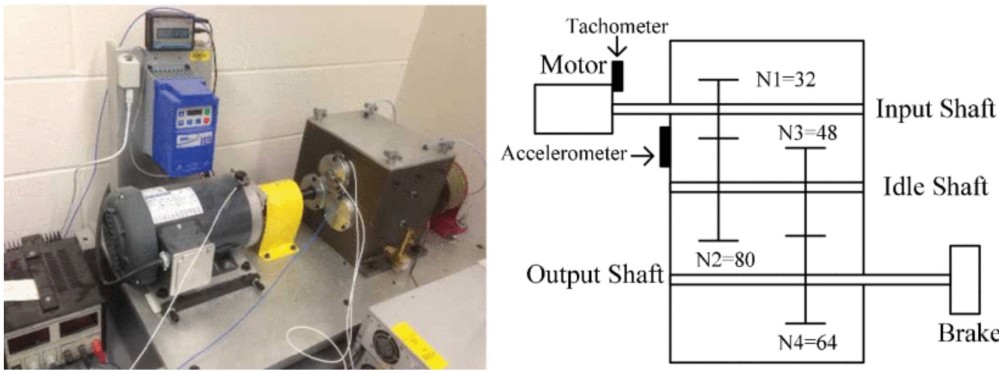

**Figure 6.** The benchmark two-stage gearbox.

In order to verify the effectiveness and reliability of the data augmentation method, we designed several datasets based on the UConn dataset. A, T1, and T1' are the original signal datasets used for training and testing. B, C, D, B', C', and D' are the datasets derived by SMOTE, DCGAN-GP, and MS-PGAN, respectively.

Table 4 represents the details of the experimental datasets from Case 1. Dataset A is an imbalanced set of data, which is processed at an imbalanced ratio of 0.1, including 312 normal samples and 32 fault samples in the other 8, totaling 568 samples. Dataset A comprises the training data outputs generated by dataset B by SMOTE, the generated dataset C by DCGAN-GP, and the generated dataset D by MS-PGAN. This expands 280 samples in each fault category into an imbalanced dataset to restore the balance of the dataset. Specifically, these datasets contain data at three scales for experimental comparison. The pure generated datasets, B', C', and D', are composed of generated data in the rebalanced datasets of B, C, and D, respectively, which are the difference sets of the two types of datasets, including 280 generated samples of 8 fault categories in each scale.

#### 4.1.2. Case 2: CWRU Dataset

In order to verify the overall performance of the bearing fault diagnosis with the proposed method using imbalanced data under actual working conditions, Case 2 selects the Case Western Reserve University (CWRU) bearing dataset [44], which is the authoritative dataset in this field. It is the standard bearing dataset published by the database of the CWRU bearing center website, which was collected from the test platform composed of a motor, torque sensor, power meter, and electronic controller. The bearing fault damage is caused by single-point damage processed by an electric spark. The drive end bearing is

selected as SKF6205, and the sampling frequency is 12 kHz for the bearing to be tested. The diameter of fault damage of the inner ring, outer ring, and rolling body of the bearing is 0.1778 mm, 0.3556 mm, 0.5334 mm, and 0.7112 mm, respectively.

**Table 4.** The details of experimental datasets, based on Case 1.

| State | Location | A | B | C | D | T1 | B′ | C′ | D′ | T1′ |
|---|---|---|---|---|---|---|---|---|---|---|
| 0 | Normal | 312 | 312 | 312 | 312 | 104 | - | - | - | - |
| 1 | Missing Tooth | 32 | 32 + 280 | 32 + 280 | 32 + 280 | 104 | 280 | 280 | 280 | 104 |
| 2 | Root Crack | 32 | 32 + 280 | 32 + 280 | 32 + 280 | 104 | 280 | 280 | 280 | 104 |
| 3 | Spalling | 32 | 32 + 280 | 32 + 280 | 32 + 280 | 104 | 280 | 280 | 280 | 104 |
| 4 | Chipping 5a | 32 | 32 + 280 | 32 + 280 | 32 + 280 | 104 | 280 | 280 | 280 | 104 |
| 5 | Chipping 4a | 32 | 32 + 280 | 32 + 280 | 32 + 280 | 104 | 280 | 280 | 280 | 104 |
| 6 | Chipping 3a | 32 | 32 + 280 | 32 + 280 | 32 + 280 | 104 | 280 | 280 | 280 | 104 |
| 7 | Chipping 2a | 32 | 32 + 280 | 32 + 280 | 32 + 280 | 104 | 280 | 280 | 280 | 104 |
| 8 | Chipping 1a | 32 | 32 + 280 | 32 + 280 | 32 + 280 | 104 | 280 | 280 | 280 | 104 |

In Table 5, datasets E, F, E′, F′, and T2 are designed based on Case 2, with ten different healthy states. We chose 0.1 and 0.05 as the imbalance ratios to verify and compare the impact of models under severe imbalance ratios. E is an imbalanced dataset with an imbalance ratio of 0.1, which comprises 840 normal samples and 84 samples for the other 9 fault categories, totaling 1596 samples. F is an imbalanced dataset with an imbalance ratio of 0.05, which has 840 normal samples and 44 samples for the other 9 fault categories, totaling 1236 samples. MS-PGAN uses the imbalanced datasets E and F as training data to obtain the multi-scale rebalanced datasets E′ and F′, including 840 samples in all categories, totaling 8400 samples. Dataset T2 is a test set with 360 samples in each category and a total of 3600 samples.

**Table 5.** The details of the experimental datasets, based on Case 2.

| State | Location | Degree (mm) | E | F | E′ | F′ | T2 |
|---|---|---|---|---|---|---|---|
| 0 | Normal | 0.000 | 840 | 840 | 840 | 840 | 360 |
| 1 | Ball | 0.1778 | 84 | 44 | 84 + 756 | 44 + 796 | 360 |
| 2 | Inner race | 0.1778 | 84 | 44 | 84 + 756 | 44 + 796 | 360 |
| 3 | Outer race | 0.1778 | 84 | 44 | 84 + 756 | 44 + 796 | 360 |
| 4 | Ball | 0.3556 | 84 | 44 | 84 + 756 | 44 + 796 | 360 |
| 5 | Inner race | 0.3556 | 84 | 44 | 84 + 756 | 44 + 796 | 360 |
| 6 | Outer race | 0.3556 | 84 | 44 | 84 + 756 | 44 + 796 | 360 |
| 7 | Ball | 0.5334 | 84 | 44 | 84 + 756 | 44 + 796 | 360 |
| 8 | Inner race | 0.5334 | 84 | 44 | 84 + 756 | 44 + 796 | 360 |
| 9 | Outer race | 0.5334 | 84 | 44 | 84 + 756 | 44 + 796 | 360 |

### 4.1.3. Data Preprocessing

Compared with the original time-domain signal, the frequency spectrum signal has more significant physical information and contains more useful information about fault diagnosis, which is helpful for quantitative analysis of the vibration signal [45]. However, traditional time-domain analysis and frequency-domain analysis are used to process vibration data that is less affected by noise or that has a simpler vibration signal. Under actual working conditions, the vibration signals of a rolling bearing may have strong non-linearity and non-stationarity, so that, when the above two signal processing methods fail to meet the requirements, the time-frequency analysis method is needed. Therefore, when considering the complex signal issues of practical engineering problems, the experiment uses variational mode decomposition (VMD) in the time-frequency analysis method to obtain frequency-domain signals as data samples to train the model. The frequency spectrum signals of the common four health states of rolling bearings after VMD processing are shown in Figure 7a–d.

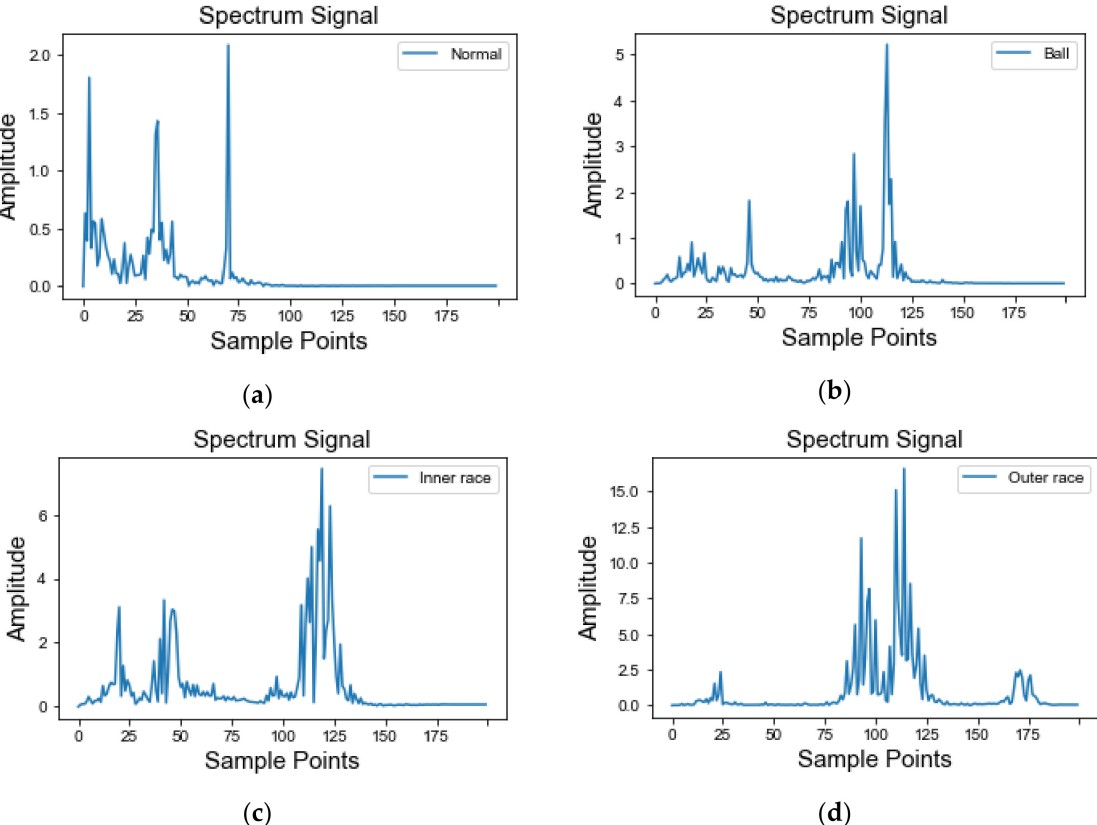

**Figure 7.** Spectrum signals of different health states, processed by VMD: (**a**) normal state; (**b**) ball fault; (**c**) inner race fault; (**d**) outer race fault.

### *4.2. Stage 1: Data Augmentation*

In order to evaluate the feasibility of the proposed two-stage fault diagnosis method to solve the imbalance problem of a vibration signal dataset of rotating machinery in application scenarios, two groups of experiments were designed to test data augmentation and fault diagnosis, respectively.

#### 4.2.1. Experiments Results of Data Augmentation

In the experiment, datasets A, E, and F are processed using a multi-scale mechanism; then, multi-scale sample data with 40-length, 100-length, and 200-length granularity are gradually generated using a multi-scale MS-PGAN. In particular, the experimental samples are randomly cut from various original signals with a length of 400 time-series segments, and the first half of the symmetric frequency domain signals are taken after VMD processing, i.e., frequency-domain signals with a length of 200.

The original frequency-domain signal diagrams of the imbalanced dataset E at 40, 100, and 200 scales are shown in Figure 8a–c, respectively. Take dataset E, for example: the multi-scale unbalanced fault samples with an imbalance ratio of 0.1 in E are iteratively trained by the gradually growing multi-scale generation model (MS-PGAN). When the training reaches Nash equilibrium, a specified number of multi-scale generated samples are generated for each fault category, and 756 samples are generated for each category at each scale, mixing with E to obtain a balanced dataset, E', containing multi-scale samples.

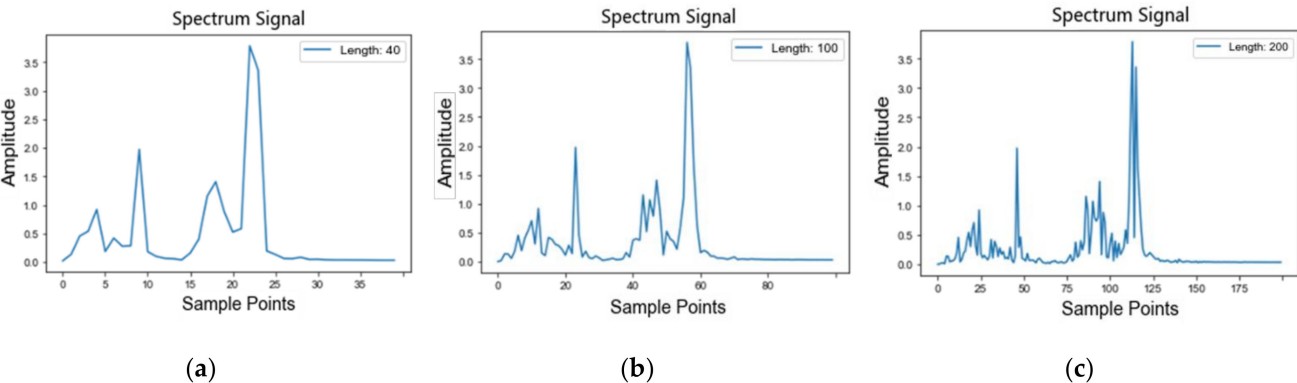

(**a**)        (**b**)        (**c**)

**Figure 8.** The spectrum of the multi-scale original signals: (**a**) low-scale; (**b**) mesoscale; (**c**) high-scale.

The frequency-domain signal maps of the generated balanced dataset E' at 40, 100, and 200 scales are shown in Figure 9a–c, respectively. In the same way, the imbalanced datasets A and F are used to obtain the corresponding multi-scale balanced generated sample datasets C and F', respectively. Eventually, these multi-scale balanced datasets will realize fault diagnosis through the MACNN-BiLSTM model of the multi-scale attention mechanism. The experimental results illustrate that MS-PGAN can generate realistic multi-scale frequency spectral signals and accord with the physical mechanism of signals, which can effectively alleviate the problem of data imbalance.

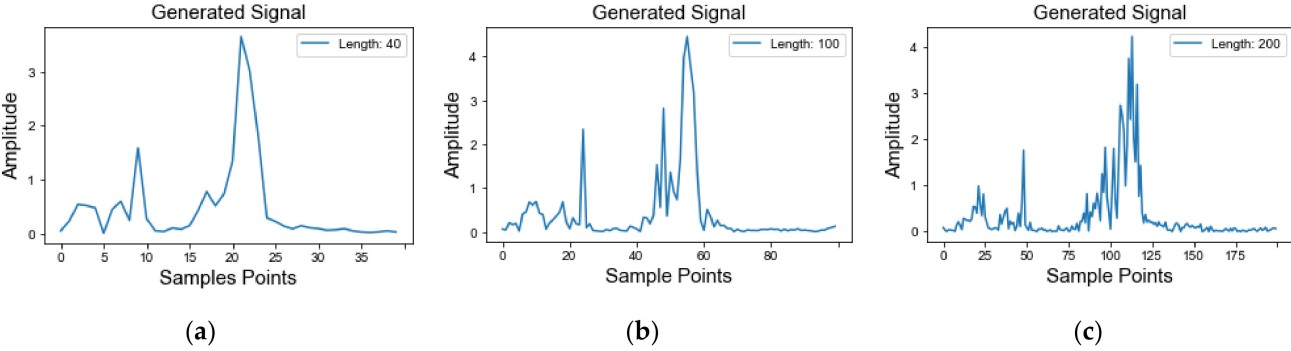

(**a**)        (**b**)        (**c**)

**Figure 9.** The spectrum of the multi-scale generated signal: (**a**) low-scale; (**b**) mesoscale; (**c**) high-scale.

### 4.2.2. Performance Analysis

Figure 10a,b illustrates the difference in ball faults of the MS-PGAN model before and after using the improved loss function MMD-WGP. The results clarify that the data generated without the improved loss function MMD-WGP has a significant problem of random spectral noise. Obviously, the presence of noise in all frequency bands disrupts the physical mechanism of the generated signal. In contrast, the generated signals with an improved MMD-WGP loss function can significantly suppress the noise problem, improve the mechanism of the generated signals, and retain more effective physical information. This indicates that the introduction of a transfer learning mechanism by MMD-WGP helps the generator model to learn a more generalized distribution of fault samples from normal samples, to avoid the limitations of learning the distribution of a few fault samples directly, which can effectively improve the stability and effectiveness of the model and restrain the GAN mode collapse problem.

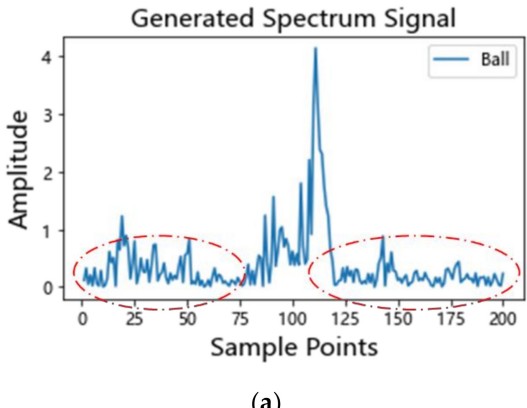 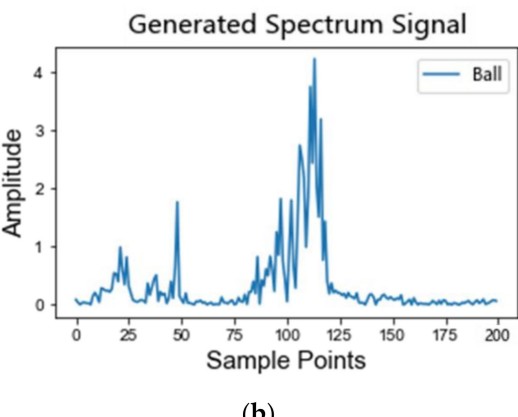

(**a**) (**b**)

**Figure 10.** The comparison of MS-PGAN using the improved loss function MMD-WGP: (**a**) generated signal without MMD-WGP; (**b**) generated signal with MMD-WGP.

This experiment demonstrates that the higher-scale fault frequency domain signal obtained by local noise interpolation upsampling not only maintains the low dimension fault feature frequency but also increases the diversity of local details by local adaptive Gaussian noise. It also greatly improves the convergence rate of the MS-PGAN model.

As shown in Figure 11a,b, when *a* is 1.5 and *b* is 1.20 in Formula (10), the fault sample of the outer race is transformed from a 40-length low-scale fault sample to a 100-length mesoscale sample, using a local noise interpolation upsampling method. Obviously, the mesoscale sample maintains the frequency of fault feature in the low-scale sample, while adding adaptive noise to improve the diversity of the generated sample.

Figure 11c,d illustrates the loss convergence without local noise interpolation for each fault class and loss convergence after using local noise interpolation during the training process for generating dataset E', respectively. The experimental results explain that loss convergence can be significantly improved by adding local noise interpolation during the progressive generation process of MS-PGAN.

As shown in Tables 6 and 7, we used the SVM model as a diagnostic model, which has outstanding stability, to conduct experiments regarding diagnostic classification on the imbalance dataset A, rebalance datasets B, C, and D, and pure generated datasets B', C', and D' at multiple scales. It can be observed that the performance of the data-driven fault diagnosis methods is prominently influenced by the imbalanced training data, and it can be effectively alleviated by expanding the dataset through a data enhancement method. In terms of the whole, the average accuracy at all scales of the pure generated datasets B', C', and D' was about 0.58%, 1.43%, and 2.50% higher than that of the imbalanced dataset A. The average accuracy at all scales of the generated rebalanced datasets B, C, and D was about 1.15%, 2.32%, and 3.05% higher than that of the imbalanced dataset A. In terms of multiple scales, there are significant differences in the classification accuracy of the models at different scales. In the three different scales, the MS-PGAN-generated rebalanced dataset D was about 3.20%, 3.43%, and 2.5% higher than imbalance set A, while the DCGAN-GP dataset was about 2.36%, 3.05%, and 1.53% higher, and the SMOTE dataset also increased by about 0.26%, 2.22%, and 0.98%. The experimental results indicate that the rebalanced dataset with generated data achieves a better classification performance than the imbalanced dataset, which proves the feasibility and usefulness of using generated data for data augmentation. Secondly, on these datasets, the quality of the MS-PGAN-generated dataset is obviously better than that of the three other datasets for all scales. In addition, the difference between the different scales of the generated data indicates that the robustness and generalization of the single-scale model are obviously lower than that of the multi-scale model, which proves the necessity of fusing multi-scale data. In conclusion, the experiments based on Case 1 demonstrate the reliability and validity of the sample quality generated via MS-PGAN.

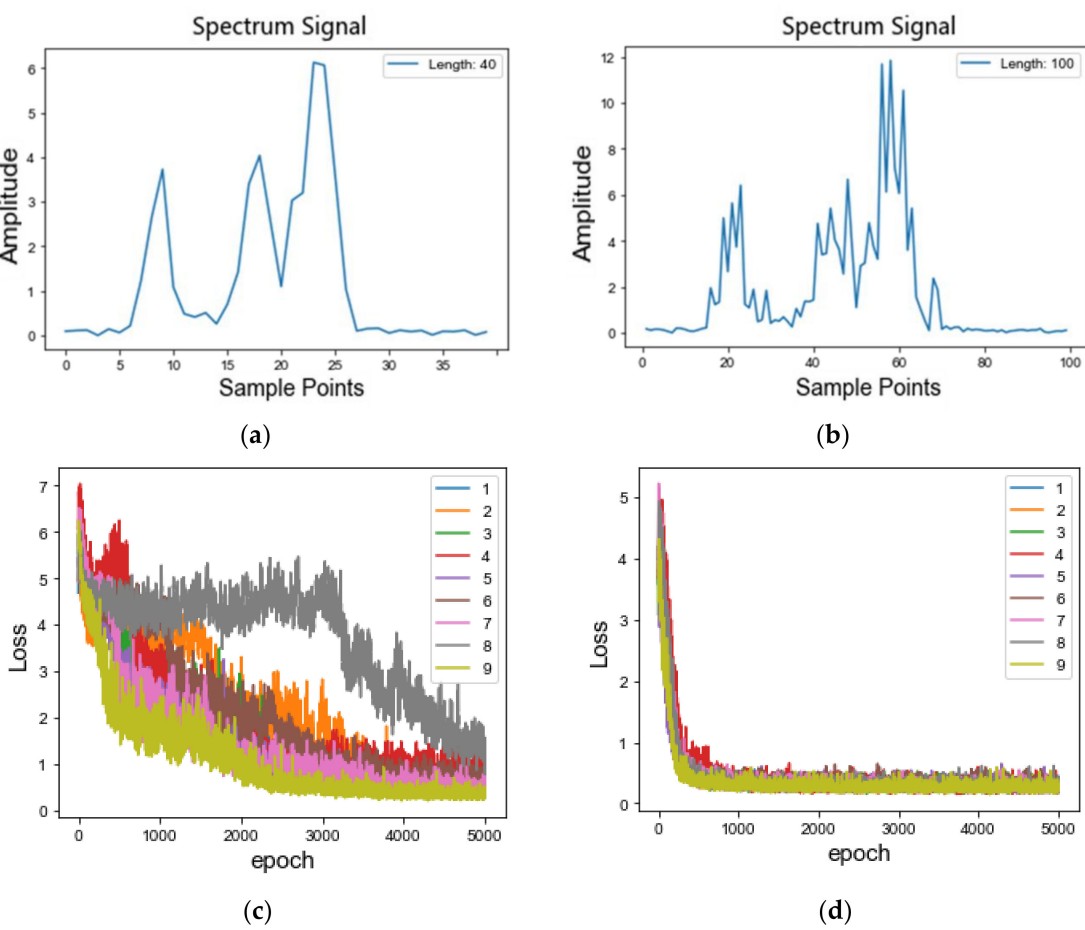

**Figure 11.** Comparison of the local noise interpolation used in the progressive generation: (**a**) a signal before using local noise interpolation; (**b**) a signal after using local noise interpolation; (**c**) the loss without local noise interpolation; (**d**) the loss using the local noise interpolation.

**Table 6.** The average precision and recall (%) of the imbalanced and rebalanced datasets by SVM.

| Data Scale | Imbalance Original Dataset A | | SMOTE Dataset B | | Rebalance DCGAN-GP Dataset C | | MS-PGAN Dataset D | |
|---|---|---|---|---|---|---|---|---|
| | aPre | aRec | aPre | aRec | aPre | aRec | aPre | aRec |
| Low | 88.48 | 88.46 | 88.74 | 88.35 | 90.84 | 90.59 | 91.68 | 91.67 |
| Middle | 90.93 | 90.60 | 93.13 | 93.06 | 93.98 | 93.91 | 94.36 | 94.34 |
| High | 92.79 | 92.84 | 93.77 | 93.91 | 94.32 | 94.34 | 95.29 | 95.19 |

**Table 7.** The average precision and recall (%) of the imbalanced and pure generated datasets by SVM.

| Data Scale | Imbalance Original Dataset A | | SMOTE Dataset B′ | | Pure Generated Data DCGAN-GP Dataset C′ | | MS-PGAN Dataset D′ | |
|---|---|---|---|---|---|---|---|---|
| | aPre | aRec | aPre | aRec | aPre | aRec | aPre | aRec |
| Low | 88.48 | 88.46 | 87.91 | 87.14 | 88.74 | 88.35 | 90.73 | 90.50 |
| Middle | 90.93 | 90.60 | 92.37 | 92.31 | 93.58 | 93.15 | 94.18 | 93.99 |
| High | 92.79 | 92.84 | 93.65 | 93.63 | 94.12 | 94.06 | 94.80 | 94.71 |

Figure 12a–d show the confusion matrix between the imbalanced dataset A and the rebalanced datasets B, C, and D at a high scale by SVM, which may provide an explanation for the practical effect of using generated data to mitigate the problem of imbalanced data. Figure 12a shows the confusion matrix of the imbalanced dataset A, illustrating that the imbalance of fault categories has a significant negative impact on fault diagnosis. Type-5 and type-7 faults are particularly affected by a category imbalance of only 82% and 79%, respectively. Figure 12b–d demonstrate that the problem of a lack of real fault data can be significantly improved by using the data expansion method. In particular, the MS-PGAN method improves the diagnostic accuracy of type-5 faults from 82% to 90%, and that of type-7 faults from 79% to 90%, achieving remarkable results. Obviously, the proposed method using MS-PGAN for data augmentation has the ability to improve the accuracy of fault diagnosis, especially for those fault categories that are most affected by an imbalance of fault categories.

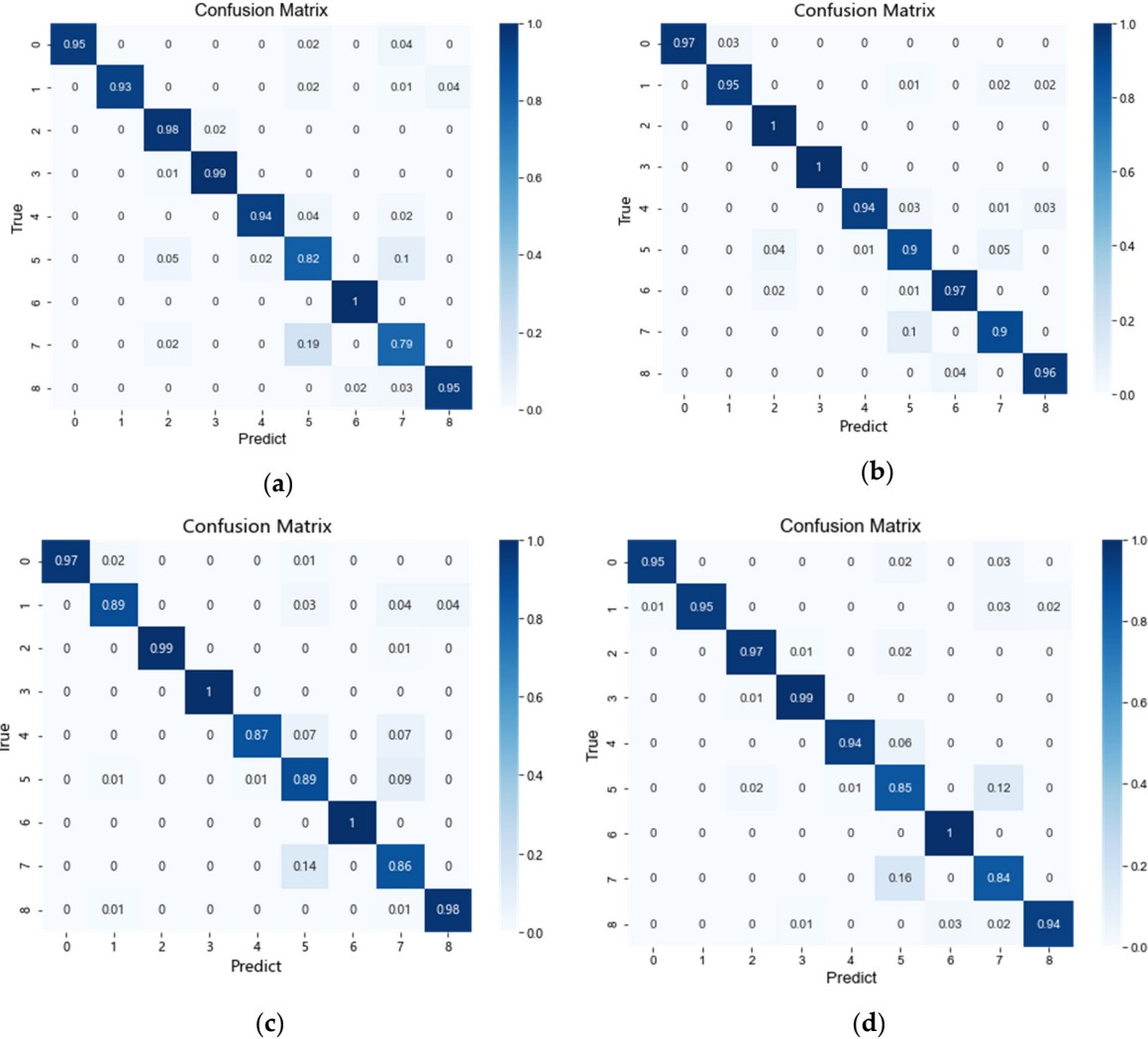

**Figure 12.** Comparison of confusion matrices under a different rebalanced dataset: (**a**) the confusion of the imbalanced dataset; (**b**) the confusion of the MS-PGAN-generated dataset; (**c**) the confusion of the DCGAN-GP-generated dataset; (**d**) the confusion of the SMOTE-generated dataset.

*4.3. Stage 2: Fault Diagnosis*

4.3.1. Experimental Results of Data Augmentation

In order to further verify the feasibility of the proposed model including MS-PGAN and MACNN-BiLSTM for fault diagnosis with imbalanced data, five baseline methods and the proposed method were selected for a comparative experiment of fault diagnosis:

VMD-SVM [8], DAE-DNN [9], 1D-CNN [10], BiLSTM [12], ResNet [11], and the proposed method. The above six methods were used to conduct comparative experiments using the datasets for Case 1 and Case 2. Table 8 shows the experimental results of the fault diagnosis using the different generation methods. Table 9 shows the experimental results of the fault diagnosis with different imbalanced ratios. In particular, all the single-scale methods used datasets at the 200-length scale to conduct the experiments.

**Table 8.** The average precision and recall (%) of baselines and our proposed method for Case 1.

| Methods | Imbalance Dataset A | | SMOTE Dataset B | | Balance DCGAN-GP Dataset C | | MS-PGAN Dataset D | |
|---|---|---|---|---|---|---|---|---|
| | aPre | aRec | aPre | aRec | aPre | aRec | aPre | aRec |
| VMD-SVM | 92.79 | 92.84 | 93.97 | 93.91 | 94.32 | 94.34 | 95.29 | 95.19 |
| SAE-DNN | 93.38 | 93.18 | 94.28 | 94.12 | 93.98 | 94.21 | 94.12 | 93.48 |
| 1D-CNN | 94.98 | 94.65 | 94.86 | 94.71 | 95.23 | 95.34 | 95.67 | 95.62 |
| Bi-LSTM | 89.54 | 89.21 | 92.74 | 92.25 | 92.41 | 92.18 | 92.92 | 92.33 |
| Res-Net | 94.44 | 93.91 | 94.97 | 94.66 | 94.87 | 94.76 | 95.30 | 95.08 |
| Ours | 95.21 | 95.13 | 95.52 | 95.24 | 96.11 | 96.02 | 97.15 | 96.89 |

**Table 9.** The average precision and recall (%) of baselines and our proposed method for Case 2.

| Methods | Imbalance 0.1 Ratio Dataset E | | 0.05 Ratio Dataset F | | Rebalance 0.1 Ratio Dataset E′ | | 0.05 Ratio Dataset F′ | |
|---|---|---|---|---|---|---|---|---|
| | aPre | aRec | aPre | aRec | aPre | aRec | aPre | aRec |
| VMD-SVM | 95.62 | 95.50 | 93.35 | 93.33 | 97.44 | 97.42 | 96.38 | 96.33 |
| SAE-DNN | 93.81 | 93.75 | 92.67 | 93.61 | 97.52 | 97.43 | 93.79 | 93.77 |
| 1D-CNN | 95.56 | 95.47 | 91.36 | 91.25 | 97.26 | 97.22 | 95.40 | 95.28 |
| Bi-LSTM | 97.35 | 97.31 | 96.26 | 96.14 | 97.57 | 97.53 | 97.53 | 97.50 |
| Res-Net | 96.56 | 96.53 | 94.92 | 95.31 | 97.22 | 97.11 | 95.92 | 95.86 |
| Ours | 97.68 | 97.59 | 96.47 | 96.35 | 98.49 | 98.47 | 98.07 | 98.03 |

4.3.2. Performance Analysis

As shown in Table 8, for all datasets based on Case 1, the average precision of the 1D-CNN, the mean of the average precision of datasets A, B, C, and D, is 95.19%, that of the BiLSTM is 91.90%, and that of the proposed model is 96.00%. In Table 9, for all datasets based on Case 2, the average precision of the 1D-CNN, the mean of the average precision of datasets E, F, E′ and F′, is 94.90%, that of the BiLSTM is 97.18%, and that of the proposed model is 97.68%. By contrast, the average precision of the BiLSTM is 2.28% higher than that of 1D-CNN in Case 2. In Case 1, the average precision of the 1D-CNN is 3.29% higher than that of the BiLSTM. The experimental results indicate that the same network structure has significantly different feature extraction effects on vibration datasets from different pieces of rotating machinery equipment, which means that the single-feature extraction unit makes the ability of fault diagnosis unstable. Therefore, in order to obtain better robustness and generalization, the diagnostic model needs a better feature extraction ability. Obviously, the multi-scale MACNN-BILSTM fault diagnosis method can achieve higher accuracy than the CNN and the BiLSTM in both Case 1 and Case 2, which proves that the proposed model can combine the advantages of the CNN and BiLSTM to extract the local and global features of the spectrum signals.

In addition, the diagnostic effect of the proposed multi-scale method is much better than that of the single-scale methods in the five experimental groups of rebalanced datasets shown in Figure 13. This proves that the diagnostic method, based on a multi-scale attention fusion mechanism, can fuse different feature information at multiple scales, and this gives the model higher accuracy and better robustness. Furthermore, as shown in Figure 14a,b, the average precision of all models in the dataset rebalanced by MS-PGAN is higher than

that in the imbalanced dataset. In particular, the proposed method achieves an average precision of 97.15%, 98.49%, and 98.07% on the rebalanced datasets C, E' and F' generated by multi-scale MS-PGAN, respectively, which is higher than the average precision of other baselines with the same imbalance ratio. This proves that the two-step multi-scale method we proposed can realize effective fault diagnosis with imbalanced data.

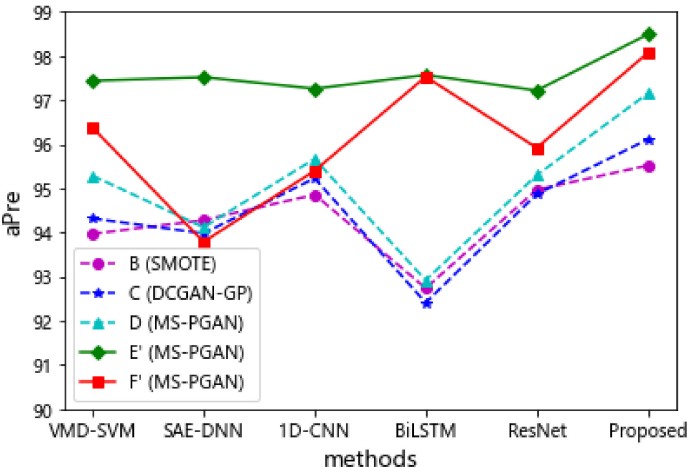

**Figure 13.** The average precision (%) of the baselines and our proposed method in all rebalanced datasets.

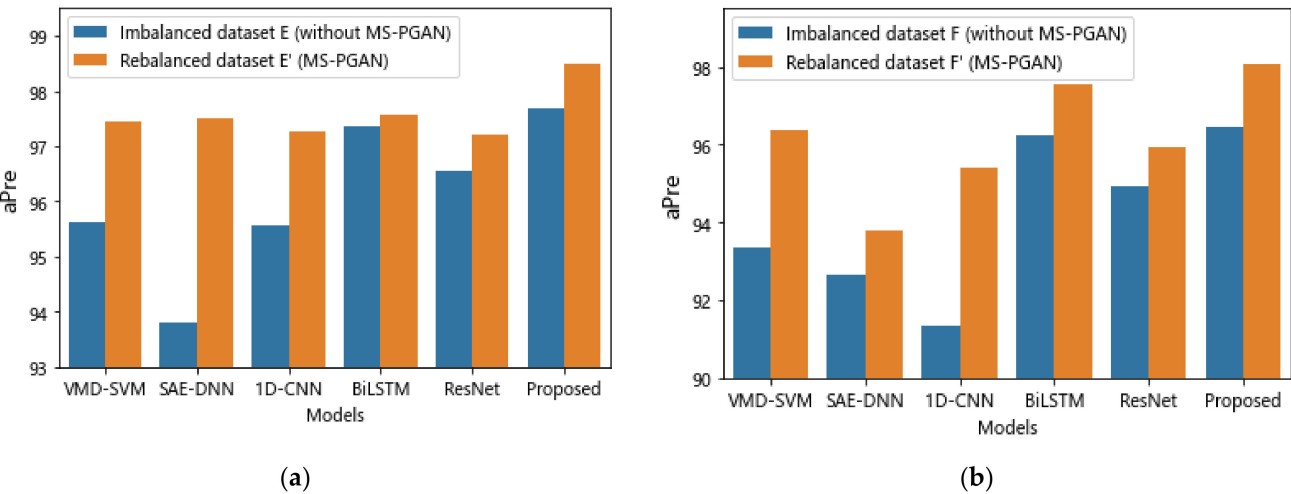

**Figure 14.** The average precision (%) of baselines and our proposed method for imbalanced data. (**a**) Results for Case 2 with a 0.1 imbalance ratio; (**b**) results for Case 2 with a 0.05 imbalance ratio.

In conclusion, the multi-scale fault diagnosis method proposed in this paper can combine the local feature extraction capability of the CNN and the global temporal feature extraction capability of the BiLSTM. It can effectively fuse the feature information at different scales through the multi-scale attention fusion mechanism, which has outstanding accuracy and robustness for different working conditions. This method achieves great generalization performance on two different datasets of rotating machinery, which has an important application value for fault diagnosis.

## 5. Conclusions

In this paper, a two-step multi-scale bearing fault diagnosis method is proposed to solve the problem of imbalanced data. In stage one, we proposed a multi-scale progressive generative adversarial network (MS-PGAN), generating high-scale samples gradually and stably from low-scale samples by means of progressive growth. In the training process, the model uses the transfer learning mechanism to learn the distribution mapping relationship

from normal samples to fault samples, which alleviates the problem of random spectral noise and mode collapse. Furthermore, local noise interpolation upsampling is used to protect the fault feature frequency and improve the convergence speed. In stage two, a diagnosis model based on a multi-scale attention mechanism (MACNN-BiLSTM) is proposed, which can extract and fuse the local frequency features and global temporal features effectively from multi-scale spectrum signals, to realize fault diagnosis. The experimental results, based on UConn and CWRU datasets, demonstrate that the proposed model can stably generate fault samples to significantly improve the imbalance problem, and can fuse more feature information at different scales with a multi-scale attention mechanism, which gives better classification accuracy, robustness, and generalization than the other compared methods.

Despite the fact that diagnostic accuracy is significantly improved after data expansion, it is still difficult to reach the upper limit of that accuracy with enough real data. This means that there is some difference between the distribution of generated samples and real samples, which requires further improvement of the generalization process. In addition, the application of the model to different pieces of rotating machinery involves more cross-domain adaptive problems. Therefore, our future work will focus on the fault diagnosis method combined with domain adaptation and domain generalization.

**Author Contributions:** Conceptualization, M.Z. and J.M.; methodology, M.Z. and Q.C.; software, M.Z. and Q.C.; validation, M.Z., Q.C. and Y.S.; formal analysis, M.Z. and Q.C.; investigation, J.M. and M.Z.; resources, J.M., Y.L.; data curation, M.Z., Q.C. and Y.L.; writing—original draft preparation, M.Z. and Q.C.; writing—review and editing, J.M. and Y.S.; visualization, M.Z.; supervision, J.M.; project administration, M.Z. and J.M.; funding acquisition, J.M. All authors have read and agreed to the published version of the manuscript.

**Funding:** This research was funded by the China Natural Science Foundation, grant number 61871432, the Natural Science Foundation of Hunan Province, grant numbers 2020JJ4275 and 2021JJ50049.

**Institutional Review Board Statement:** Not applicable.

**Informed Consent Statement:** Not applicable.

**Data Availability Statement:** The data sets generated and/or analyzed during the current study are available from the corresponding author on reasonable request.

**Conflicts of Interest:** The authors declare no conflict of interest.

## Abbreviations

| | |
|---|---|
| 1D-CNN | One-dimensional Convolutional Neural Network |
| 1D-Conv | One-dimensional Convolution Layer |
| 1D-ConvT | One-dimensional Transposed Convolution Layer |
| BiLSTM | Bidirectional Long Short-Term Memory Network |
| CNN | Convolutional Neural Network |
| DNN | Deep Neural Network |
| DCGAN | Deep Convolutional Generative Adversarial Network |
| IMF | Intrinsic Mode Function |
| GAN | Generative Adversarial Network |
| LSTM | Long Short-Term Memory Network |
| MS-PGAN | Multi-scale Progressive Generative Adversarial Network |
| MACNN-BiLSTM | Multi-scale Attention CNN-BiLSTM |
| MMD | Maximum Mean Discrepancy |
| ResNet | Deep Residual Network |
| ReLU | Rectified Linear Units |
| SAE | Stack Auto Encoder |
| SVM | Support Vector Machine |
| Tanh | Hyperbolic Tangent |
| VMD | Variational Mode Decomposition |

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
