# Peer review of "Two-Stage Multi-Scale Fault Diagnosis Method for Rolling Bearings with Imbalanced Data"

_machines, doi:10.3390/machines10050336_

Round 1

Reviewer 1 Report

The English of your manuscript must be improved. In other words, your manuscript needs careful editing by someone with expertise in technical English. When editing, please pay particular attention to English grammar, and sentence structure throughout the manuscript. The following queries  needs to be addressed

  • List of abbreviations to be included in the manuscript
  • Table 4,  Mismatch in the number of samples, Total 9 defective states as mentioned in the article, only 8  are shown. If you go by this also, total number of samples is 568 but it  is given as 586, verify and correct this.
  • In table 5, If you consider 9 defective conditions, then total number of samples is 600. But in the article it is mentioned as 586 which is wrong I think.
  • In table 5, Imbalance ratio of case 2 data set for E and F is  (0.1 and 0.05 is different any reasons.
  • Introduction section can be reduced by concisely witting reviews  14,15,17,19 and 34. Also, summary of the contribution needs to be reduced.
  • Data set G is missing in table5
  • In sec., 4.2.2,  3rd para, when a is 1.5 and b is 1.2 –but In equation 9, a and b terms are missing. Carefully check the all the equations and appropriately use in the article

Author Response

Point 1: List of abbreviations to be included in the manuscript.

Response 1: Thanks for your reviews. We have added abbreviations in the revised manuscript.

Point 2: Table 4, Mismatch in the number of samples, Total 9 defective states as mentioned in the article, only 8 are shown. If you go by this also, total number of samples is 568 but it  is given as 586, verify and correct this.

Response 2: As you pointed out, there are actually 8 defect states , total number of samples is 568.

Point 3: In table 5, If you consider 9 defective conditions, then total number of samples is 600. But in the article it is mentioned as 586 which is wrong I think.

Response 3: As you pointed out, there are actually 8 defect states , total number of samples is 568.

Point 4: In table 5, Imbalance ratio of case 2 data set for E and F is  (0.1 and 0.05 is different any reasons.

Response 4: In case2, we selected two imbalance ratios of different degrees, 0.1 and 0.05, to verify and compare the impact of methods under severe imbalanced data.

Point 5: Introduction section can be reduced by concisely witting reviews  14,15,17,19 and 34. Also, summary of the contribution needs to be reduced.

Response 5: Thanks for your reviews. We have reduced the introduction and contribution in the revised manuscript.

Point 6: Data set G is missing in table5.

Response 6: As you pointed out, this is a writing error, it should actually be dataset T2.

Point 7: In sec., 4.2.2,  3rd para, when a is 1.5 and b is 1.2 –but In equation 9, a and b terms are missing. Carefully check the all the equations and appropriately use in the article.

Response 7: As you pointed out, this is a writing error, it should actually be equation 10.

Reviewer 2 Report

  • Table 1, 2 – it is worth to say about what 1D-Conv and 1D-ConvT are.
  • Right after the formula (10) it is written that ? value determines the local window size. But we don’t have ? in (10).
  • It is also necessary to match the numbers indicated in line 450,451 on the size of the defect with similar numbers in Table 5. In both cases, millimeters are indicated, but the numbers are different. Perhaps in the second case you have inches.
  • The improvement in precision and recall when using the proposed MS-PGAN method compared to the already existing SMOTE, DCGAN-GP is about 1%, according to Tables 6,7,8. Does it really matter in practice? It would be nice to compare the methods in terms of computational complexity and time.
  • Line 574 – check the number 87%, it does not match what we see in the Fig. 12b.
  • I don’t see the average precision values (that the authors write about in lines 594-599) in tables 8,9.
  • Typos and English need to be corrected, for example in Line 329 second ‘the’ is odd, in formulas (8), (9), f(y_i) has an extra parenthesis. In Line 541 we should have ‘rebalanced’ instead of ‘rebalance’, see also ‘imbalance’ in Line 540.

Author Response

Point 1: Table 1, 2 – it is worth to say about what 1D-Conv and 1D-ConvT are.

Response 1: Thanks for your reviews. Convolution in deep learning is usually used to process two-dimensional images. To process one-dimensional signal data, we use 1D-ConV and 1D-convt. 1D-conv refers to one-dimensional convolution layer, which is used to extract and compress one-dimensional input features. 1D-convt refers to one-dimensional transposed convolution layer, which is used to amplify the length of one-dimensional input data. It works almost exactly the same as one-dimensional convolutional layer, but in reverse. We have added the explanation of 1D-conv and 1D-convt in the revised manuscript.

Point 2: Right after the formula (10) it is written that ? value determines the local window size. But we don’t have ? in (10).

Response 2: As you pointed out, the  value is not in Formula 10, but determines the number of local Windows according to the value of . We have added details in the revised manuscript.

Point 3: It is also necessary to match the numbers indicated in line 450,451 on the size of the defect with similar numbers in Table 5. In both cases, millimeters are indicated, but the numbers are different. Perhaps in the second case you have inches.

Response 3: As you pointed out, the unit of Degree in Table 5 is actually inches. We have unified the unit of measurement in the revised manuscript.

Point 4: The improvement in precision and recall when using the proposed MS-PGAN method compared to the already existing SMOTE, DCGAN-GP is about 1%, according to Tables 6,7,8. Does it really matter in practice? It would be nice to compare the methods in terms of computational complexity and time.

Response 4: Thanks for your reviews. Several explanations about MS-PGAN mothed is following:

In terms of performance, MS-PGAN is designed to address the problem of imbalanced data. As shown in Figure 12, in the categories most affected by imbalanced data, type 5 and 7, MS-PGAN improves accuracy by 8% and 11%, 1% and 4% higher than DCGAN-GP and 5% and 6% higher than SMOTE. Obviously, DCGAN-GP performed better and SMOTE performed worse. However, not all fault identification is affected by imbalanced data in the experiment, such as type 2 and 3 in Figure 12, where Data Augmentation methods have little effect. Therefore, after averaging, the overall precision and recall of MS-PGAN seem to have increased by about 1%. In fact, MS-PGAN is more effective in solving the problem of imbalanced data. Further, the key multi-scale mechanism of MS-PGAN Combining MACNN-BiLSTM can extract and fuse features from different scales. In addition, the quality of generated data is also a key issue in practice. MS-PGAN improves the spectral noise of generated data and protects mechanism information, while DCGAN-GP does not.

In terms of convergence, as shown in Figure 11c-d, the convergence speed of MS-PGAN with Local Noise Interpolation Upsampling is greatly improved, which is about 10 times faster than that without it.

In terms of availability, the structure of MS-PGAN structure is very stable and avoids training failure and mode collapse for one-dimensional vibration signal, which DCGAN-GP does not.

In practice, since the method we proposed is a separable two-stage method, we can use MS-PGAN for categories which greatly affected by imbalanced data, and flexibly combine data augmentation method and fault diagnosis method.

Point 5: Line 574 – check the number 87%, it does not match what we see in the Fig. 12b..

Response 5: As you pointed out, this is a writing error and actually be 90%, as we can see in Figure 12b.

Point 6: I don’t see the average precision values (that the authors write about in lines 594-599) in tables 8,9.

Response 6: This average accuracy (about in lines 594-599) is calculated from table 8,9. For example, in Case1, the average precision of 1d-CNN is actually the mean of the average precision under four conditions (dataset A, B, C and D). The calculation process is . We have added explanations and corrected calculation errors in the revised manuscript.

Point 7: Typos and English need to be corrected, for example in Line 329 second ‘the’ is odd, in formulas (8), (9), f(y_i) has an extra parenthesis. In Line 541 we should have ‘rebalanced’ instead of ‘rebalance’, see also ‘imbalance’ in Line 540.

Response 7: Thank you for your reviews. We have corrected these spelling and grammar errors in the revised manuscript.
